# Prediction and Optimization of Electrospun Polyacrylonitrile Fiber Diameter Based on Grey System Theory

**DOI:** 10.3390/ma12142237

**Published:** 2019-07-11

**Authors:** Qihong Zhou, Liqun Lin, Ge Chen, Zhaoqun Du

**Affiliations:** 1College of Mechanical Engineering, Donghua University, Shanghai 201620, China; 2College of Textiles, Donghua University, Shanghai 201620, China

**Keywords:** electrospinning, nanofiber diameter prediction, grey prediction theory, background value optimization, fractional order accumulation

## Abstract

This paper provides a new method for predicting the diameter of electrospun nanofibers. Based on the grey system theory, the effects of polyacrylonitrile mass fraction, voltage, flow rate, and receiving distance on fiber diameter were studied. The GM(1,1) (grey model) model and DNGM(1,1) (The DNGM (1,1) model is based on the whitening differential equation using parameters to Directly estimate the approximate Non-homogeneous sequence Grey prediction Model) model were established to predict fiber diameter by a single-factor change, and the results showed high prediction accuracy. The multi-variable grey model MGM(1,n) (MGM(1,n) is a Multivariate Grey prediction Model) was used for prediction of fiber diameter when multiple factors change simultaneously. The results showed that the average modeling fitting error is 8.62%. The background value coefficients of the MGM(1,n) model were optimized, the average modeling fitting error was reduced to 1.01%, and the average prediction error was reduced to 1.33%. Combining the fractional optimization with the background-value coefficient optimization, the optimal background-value coefficient α and the order r were selected. The results showed that the average modeling fitting error is 0.85%, and the average prediction error is 0.38%. The results demonstrate that the grey system theory can effectively predict the diameter of polyacrylonitrile electrospinning fibers with high prediction accuracy. This theory can increase the control of nanofiber diameters in production.

## 1. Introduction

Nanofiber membranes prepared via electrospinning have extremely high porosity, high surface-to-volume ratios [1], and excellent surface properties leading to broad application prospects in textiles, filtration, medicine, sensing, and other fields [2,3,4,5,6,7]. Its properties also depend on different electrospun parameters. The fiber diameter seriously affects the pore size per unit area and affects the specific area of the fiber membrane, thus affecting the filtration and gas permeability of the fiber membrane [8]. Therefore, the prediction of fiber diameter and its control is of great significance for the performance and improvement of nanofiber membranes [9].

At present, two methods are used for predicting and controlling the diameter of electrospun nanofibers. One is to study the relationship between the jet radius and its moving distance in the direction of stretching [10]. The other is to study the relationship between fiber diameter and process parameters [11,12,13].

Spivak [10] comprehensively studied the factors of inertial force, hydrostatic force, viscous resistance, electric-field force, and surface tension. The relationship scaled between the jet-flow diameter and its running distance. However, the jet-flow radius varies with the electrospinning-process parameters and the properties of the solution; thus, the relationship is not satisfactory in terms of universality.

Yarin [14] established an abstract mathematical model for the charged liquid bead between the spinneret and the collector. The model mainly conducts theoretical analysis and does not combine the experimental data for full analysis and verification. Stepanyan et al. [15,16] believed that the fiber diameter is caused by the combination of the charge repulsive force on the fiber and the viscous force after evaporation of the solvent. The charge repulsion on the fibers causes the fibers to stretch, thereby making the fiber diameter smaller. The viscous resistance after evaporation of the solvent prevents the fiber from stretching. During the electrospinning process, a longer solvent volatilization time and stretching time will lead to the jet being stretched, and thus, a finer fiber. Assuming that the solvent is completely volatilized and fully stretched, one can derive the relationship of the fiber diameter under ideal conditions. Since this relationship is based on the assumption that the solvent is completely volatilized, it is not very suitable for actual production.

Wei [17] studied the effects of carboxymethyl chitosan solution concentration and electrospinning process parameters on the diameter of nanofibers. She selected four factors including solution concentration, voltage, flow rate, and nozzle inner diameter. The Box–Behnken design in the response-surface method was used to predict the diameter of carboxymethyl chitosan electrospinning fiber, and a quadratic regression model was obtained [18,19,20,21,22]. The results show that the model determines the coefficient R^2^ to be 0.9138, which indicates that the prediction accuracy of the fiber diameter is 91.38%. Although the prediction effect is better than Hou’s research results, he used response-surface methodology to predict the diameter of fibers with an accuracy of 86.59% [23], the prediction error is still large.

The grey prediction theory mainly studies some systems where some of the information is known and some of the information is unknown. Grey theory is suitable for the prediction of small sample data. The theory makes the system characteristics strengthen by processing the known information, so that the law of system operation can be correctly recognized. The goal of objective evaluation and prediction of the system is finally achieved [24,25]. At the same time, it also provides strong support for the subsequent decision-making of the system. The grey prediction theory can not only predict unknown data, but also obtain continuous time series by processing discrete time series to realize the prediction of incomplete information in the system [26].

Electrospinning to prepare fibers has many factors affecting fiber diameter, some factors are known, and some factors are unknown. At present, several factors that have been recognized as the most influential factors are mass fraction, voltage, flow rate, and receiving distance. Grey theory prediction is a method for predicting systems with uncertainties. The grey theory scientifically evaluates the research object by modeling and predicting the main factors of the research object. Moreover, grey theory is very suitable for small sample prediction, which reduces the cost of the experiment and improves the efficiency of the experiment. In summary, grey theory is very suitable for the prediction of the diameter of electrospun nanofibers.

Grey theory uses grey-correlation analysis to analyze the system. For a system with development and change, the correlation analysis is actually a quantitative analysis of the development trend of the system [27]. GM(1,1) is the most basic univariate prediction model in grey theory. The model is mature and extensive. When a single variable rises to a multivariate, the multi-variable grey prediction model GM(1,n) makes a forced linear assumption between the parameters, and thus the prediction accuracy of the model low. The MGM(1,n) (MGM(1,n) is a Multivariate Grey prediction Model) model is derived from the GM(1,1) model under the n variables. It is not a simple combination of n GM(1,1) models; rather, it constructs n n-variable differential equations and solves them. It can better reflect the relationship between mutual constraints and mutual development among multiple variables [28,29].

In practice, the grey prediction model needs to be appropriately improved to achieve better prediction results for different research objects. With the continuous optimization of the grey prediction model, the prediction accuracy of the model for the research object is continuously improved. Model accuracy is the key and difficult point in building a model. Many researchers have proposed methods to improve accuracy often using background value reconstruction [30], background value coefficient optimization [31], initial value optimization [32], and the new information optimization model [33]. When building a grey prediction model, in order to facilitate the model solution, an accumulation sequence is often represented by a sequence of mean generations. It is common to say that the background value α is taken as 0.5. However, according to the actual situation, selecting the appropriate background value coefficient α can effectively improve the prediction accuracy.

The models mentioned above are all integer-order models, but in practice many objects satisfy the characteristics of fractional order, and the methods of fractional-order accumulation can better reveal their characteristics. Wu and Liu proposed a fractional-order grey model FGM(1,1) (FGM(1,1) is a Fractional-order Gray prediction Model), the prediction error of which can be reduced by using fractional power in grey generation [34,35].

In this paper, the principle and control of electrospinning are analyzed. The effects of main factors such as mass fraction, voltage, flow rate, and receiving distance on fiber diameter were studied, and an effective prediction model is found for the prediction of electrospinning fiber diameter. The GM(1,1) model was used to predict the diameter of single-variable electrospinning fibers. The original system sequence prediction accuracy was explored in the multivariate prediction of electrospinning fiber diameter based on the MGM(1,n) model. In the modeling principle, the method of optimizing the background value is adopted to correct the prediction fitting error caused by the selection of the background value parameters. To further improve the prediction accuracy, the fractional order idea is introduced into the modeling mechanism. The fractional order and background value optimization are combined to establish an FMGM(1,n) (FMGM(1,n) is a Fractional order Multivariate Grey prediction Model) model that is more adaptive to the system sequence.

## 2. Experimental Section

### 2.1. Materials and Instruments

The materials include polyacrylonitrile (PAN, M_w_ = 150,000 g/mol) and N, N-dimethylformamide (DMF, density = 0.945–0.950 g/mL). The electronic balance (ME104/02) was purchased from Mettler Toledo instruments co., Ltd. (Shanghai, China). A magnetic stirrer (85-2A) was purchased from the Tianjin Sardis experimental analytical instrument manufacturer (Tianjin, China). An environmental scanning electron microscope (ESEM, Quanta 250) manufactured by FEI Corporation (Hillsborough, OR, USA) and a laboratory custom electrospinning machine were used.

### 2.2. Preparation of PAN and Testing

The appropriate amount of DMF was measured in a beaker. The PAN powder was then dried in a low temperature oven for 2 h. The PAN powder was slowly added to the solvent and transferred to a three-necked flask. The three-necked flask was placed in a constant temperature water bath at 60 °C, and the sample was stirred with a magnetic stirrer for 4 h. The stirred solution exhibited a stable bright yellow solution. The solution was allowed to stand for a period of time for defoaming treatment, and then transferred to a sealed beaker. The beaker containing the solution was labeled.

The control-variable method was adopted to explore the effect of mass fraction factors on fiber diameter. The voltage was set to 18 kV, the distance between the nozzle and the collecting plate was set to 16 cm, the inner diameter of the nozzle was selected to be 0.57 mm, and the flow rate of the solution was set to 0.5 mL/h. PAN electrospinning has a good spinning mass fraction ranging from about 8% to 16%. When the mass fraction is low, the bead-like structure is likely to occur, and when the mass fraction is high, the spun fiber membrane is not easily formed. Therefore, the mass fraction of this experiment was selected to be in the middle of the range. The electrospinning experiment was carried out with a solution having a PAN mass fraction of 9%, 10%, 11%, 12%, 13%, 14%, 10.5%, 11.5%, and 12.5%. When studying the voltage factor, flow rate factor, and receiving distance factor, the control variable method was similarly used, and the fixed PAN mass fraction was 12%. When one factor was studied, the other three factors remained unchanged, and only the value of the factor being studied was changed. The parameter values of specific factors are detailed in Table 1.

### 2.3. Sample Detection

The fiber membrane prepared according to the above experimental conditions was placed in a petri dish and vacuum dried at a low temperature for 8 h. The nanofiber membrane was cut into a sample of appropriate size and fixed on the stage with a conductive paste. The electrospun fiber membrane was observed using an environmental scanning electron microscope (ESEM, FEI Corporation, Hillsborough, Oregon, USA). Ten fibers were randomly selected in the ESEM image, and the diameter of the fibers were measured by the image analysis function in Photoshop software (Version 13.0, Adobe Systems Software Ireland Ltd, San Jose, California, USA), and the average value was calculated as the fiber diameter of the experimental spinning. The data of the fiber diameter under specific factors are shown in Table 1.

## 3. Single-Factor Model Establishment and Experimental Verification

Under the premise of ensuring the quality of the electrospun fiber membrane, the parameter values of various factors were set within the maximum range. For example, when the loading voltage was lower than 10 kV, the fiber membrane on the collecting plate was clustered although the fiber could be spun. When the loading voltage was 10–12 kV, the ESEM showed more bead-like structures, which seriously affects the quality of the fiber membrane. The bead-like structure is shown in Figure 1. Electrospinning is unstable when the loading voltage is higher than 22 kV. From this, we determined that the normal range of the spinning voltage is 12 kV to 22 kV. Some fiber morphology ESEM images between 12 kV and 22 kV are shown in Figure 2.

The loading voltage at the nozzle was recorded as V unit (kV), the mass fraction of PAN solution was recorded as M unit (%), the flow rate of the solution at the nozzle was recorded as S unit (mL/h), the distance between the nozzle and the collecting plate was recorded as L unit (cm), and $x_{i}^{\left(0\right)}$ indicates the corresponding value of nanofiber diameter (nm). In Table 1, No. 1–6 are the parameter values of each factor, which is the maximum range value under the premise of ensuring the quality of the fiber membrane; parameter value of each factor increased sequentially. This part of the data was used to build a grey model. In grey prediction theory, the theory uses at least two sets of data to test the accuracy of the model. In this paper, three sets of data were used to test the accuracy. The data in sequence numbers 7–9 are interpolation data within the respective factors, and this part of the data was used to test the model’s prediction accuracy.

### 3.1. Grey Relational Analysis

In the grey theory, if two factors change with the development of the system, then the trend of the two factors has a high similarity indicating that the correlation between the two is higher. On the contrary, if the two factors change with the development of the system, the trend has a low similarity the degree of correlation is low [36]. The four above groups of factors underwent relational analysis to explore the correlation between various factors and fiber diameter. Taking the relational analysis of voltage factor and fiber diameter as an example, the raw data are expressed as the following:$$X^{\left(0\right)}=\left[\begin{array}{cccc} x_{1}^{\left(0\right)}(1) & x_{1}^{\left(0\right)}(2) & \cdots & x_{1}^{\left(0\right)}(M)\\ x_{2}^{\left(0\right)}(1) & x_{2}^{\left(0\right)}(2) & \cdots & x_{2}^{\left(0\right)}(M)\\ \vdots & \vdots & & \vdots \\ x_{n}^{\left(0\right)}(1) & x_{n}^{\left(0\right)}(2) & \cdots & x_{n}^{\left(0\right)}(M) \end{array}\right]=\left[\begin{array}{cccccc} 12 & 14 & 16 & 18 & 20 & 22\\ 322.5 & 342.0 & 351.4 & 353.9 & 364.8 & 368.6 \end{array}\right]$$

Step 1: The physical units between the original data sequences are different. In order to make the data sequences comparable, it is necessary to preprocess the data sequence, eliminate the order of magnitude difference and dimension of the series, and retain the trend of data changes. The raw data are processed using the mean conversion equation as shown in Equations (1) and (2).
(1)$$x_{i}^{\left(0\right)}=\frac{\sum_{t=1}^{M}x_{i}^{\left(0\right)}(t)}{M}(t=1,2,\cdots ,M)$$
(2)$$x_{i}^{\left(1\right)}(t)=x_{i}^{\left(0\right)}(t)/x_{i}^{\left(0\right)}(i=1,2,\cdots n)$$

Step 2: As shown in Equation (3), the absolute value $\Delta_{ij}(t)$ of the difference corresponding to the parameter value of each factor is solved. We then selected the maximum value Δmax and the minimum value Δmin in the absolute value and solved the correlation coefficient $L_{ij}(t)$ using Equation (4):(3)$$\Delta_{ij}(t)=\left|x_{i}^{\left(1\right)}(t)-x_{j}^{\left(1\right)}(t)\right|(j=1,2,\cdots ,n;j\neq i)$$
(4)$$L_{ij}(t)=(\Delta \min+k\Delta \max)/(\Delta_{ij}(t)+k\Delta \max)$$

In Equation (4), k takes a value of 0.5, and the calculated absolute value has a maximum value of Δmax = 0.2426 and a minimum value of Δmin = 0.0492.
$$L_{12}=\left[\begin{array}{cccccc} 0.5083 & 0.6236 & 0.9339 & 1.0000 & 0.6633 & 0.4686 \end{array}\right]$$

Step 3: The final step solves the value of the degree of grey relation:(5)$$R_{ij}=\frac{\sum_{t=1}^{M}L_{ij}(t)}{M}(t=1,2,\cdots ,M)$$

From Equation (5), the grey relational degree between the voltage and the fiber diameter is $R_{1}$ = 0.6696.

Similarly, using the above formula, the value of the grey relational degree between the other three factors and the fiber diameter is obtained. The results are as follows: the grey relational degree value of the mass fraction $R_{2}$ = 0.5670, the grey relational degree of the flow rate $R_{3}$ = 0.6631, and the grey relational degree of the distance factor $R_{4}$ = 0.6752. Analysis and comparison of the above four factors and fiber-diameter grey relational degree indicated that the grey relational degree of each factor was greater than 0.5, indicating that these four factors have a greater impact on fiber diameter. The results show that it is reasonable to select these four factors to study the diameter of nanofibers and can be used for the subsequent grey prediction theory modeling.

### 3.2. Establish GM(1,1) Model and Verification

The GM(1,1) model was established by taking the voltage factor as an example when studying a single factor. The value of the fiber diameter corresponding to the uniform change of the voltage is recorded as a one-dimensional original data sequence. Its expression form is:$$X^{\left(0\right)}=(x^{\left(0\right)}(1),x^{\left(0\right)}(2),\cdots ,x^{\left(0\right)}(m))$$

In order to get a better system of growth, random interference from the original data sequence needs to be eliminated. The raw data is subjected to an accumulation process to obtain $X^{\left(1\right)}$ sequence.
$$X^{\left(1\right)}=(x^{\left(1\right)}(1),x^{\left(1\right)}(2),\cdots ,x^{\left(1\right)}(m))$$

The accumulation process is as shown in Equation (6):(6)$$x^{\left(1\right)}(k)=\sum_{i=1}^{k}x^{\left(0\right)}(i);k=1,2,\cdots ,m$$

The background value sequence $Z^{\left(1\right)}$ is obtained by processing $X^{\left(1\right)}$ with Equation (7). Here, α is the background value weight coefficient. The value of α is in the range (0, 1), usually for the convenience of calculation, the average generation series modeling calculation is generally adopted. That is, the value of α is taken as 0.5 [37].
(7)$$z^{\left(1\right)}(k)=\alpha x^{\left(1\right)}(k-1)+(1-\alpha )x^{\left(1\right)}(k),k=2,3,\cdots m$$
$$Z^{\left(1\right)}=(z^{\left(1\right)}(2),z^{\left(1\right)}(3),\cdots ,z^{\left(1\right)}(m))$$

The whitenization Equation (8) in the GM(1,1) model is discretely processed to obtain a univariate first-order grey prediction model (Equation(9)).
(8)$$\frac{dx^{\left(1\right)}}{dt}+ax^{\left(1\right)}=b$$
(9)$$x^{\left(0\right)}(k)+az^{\left(1\right)}(k)=b$$

In Equation (9), a is the main variable parameter, and b is the background value of the model. The parameter matrix of the a and b values is determined by the least squares method. The solution is as follows:(10)$$\overset{\wedge }{a}={\left(a,b\right)}^{T}$$
(11)$$\overset{\wedge }{a}={\left(B^{T}B\right)}^{\left(-1\right)}B^{T}Y$$

In Equation (11), the matrices B and Y are written as follows:(12)$$Y=\left[\begin{array}{c} x^{\left(0\right)}(2)\\ x^{\left(0\right)}(3)\\ \vdots \\ x^{\left(0\right)}(m) \end{array}\right]B=\left[\begin{array}{cc} -z^{\left(1\right)}(2) & 1\\ -z^{\left(1\right)}(3) & 1\\ \vdots & \vdots \\ -z^{\left(1\right)}(m) & 1 \end{array}\right]$$

The solved parameters a and b are substituted into the whitenization Equation (8) to obtain the GM(1, 1) time response function Equation (13). On this basis, the discretization Equation (14) is obtained.
(13)$$x^{\left(1\right)}(t)=Ce^{-at}+\frac{b}{a}$$
(14)$${\overset{\wedge }{x}}^{\left(1\right)}(k+1)=Ce^{-at}+\frac{b}{a},k=0,1,2,\cdots ,m-1$$

Substituting the initial values $X^{\left(1\right)}=X^{\left(0\right)}$ and k=0 into Equation (14), the value of the undetermined parameter C can be obtained. On this basis, an accumulated analog value sequence ${\overset{\wedge }{X}}^{\left(1\right)}$ is obtained. The adjacent two values of the ${\overset{\wedge }{X}}^{\left(1\right)}$ sequence are subtracted as shown in Equation (15). The final reduction yields the simulated and predicted value sequence ${\overset{\wedge }{x}}^{\left(0\right)}$ corresponding to the original data.
(15)$${\overset{\wedge }{x}}^{\left(0\right)}(k+1)={\overset{\wedge }{x}}^{\left(1\right)}(k+1)-{\overset{\wedge }{x}}^{\left(1\right)}(k)$$

The model error is evaluated using the average relative error MAPE (The mean absolute percentage error (MAPE) is a measure of prediction accuracy of a forecasting method in statistics) as shown in Equation (16).
(16)$$MAPE(\%)=\frac{1}{n}\sum_{k=1}^{n}\left|\frac{x^{\left(0\right)}(k)-{\overset{\wedge }{x}}^{\left(0\right)}(k)}{x^{\left(0\right)}(k)}\right|\times 100\%$$

In order to guarantee the quality of the product, the prediction error must be within 3%. If the average relative error exceeds the target error, then a corresponding model improvement is required. The Matlab program of the GM(1,1) model can be written such that the original data sequence under each factor is brought into the program. The fiber diameter fitting value and predicted value corresponding to each factor were obtained. The results are shown in Table 2 and Table 3. The Real Value is a measure of the fiber diameter. The Simulation Value is the fitted value of the GM(1,1) model. The Simulation Error is the relative error between the fitted value and the measured value. The Forecast Value is the predicted value of the GM(1,1) model. The Forecast Error is the relative error between the predicted value and the measured value. MAPE_1 (Mean Absolute Percentage Error) is the average of the Simulation Error. MAPE_2 (Mean Absolute Percentage Error) is the average of Forecast Error. The k is the serial number.

It can be seen from Table 2 and Table 3 that the fitting error and prediction error of the voltage factor are the smallest. It is shown that the fitting value and the measured value of the voltage factor have the best fitting effect. The average relative fitting error is within 2.5% for the four factors above except for the mass fraction factor; the average relative prediction error is less than 2%. This shows that the model has higher prediction accuracy. The forecasting requirements can be met in actual production activities. The average modeling fit accuracy is 3.35% for the mass fraction factor sequence, which exceeds the expected error target. The model needs research and improvement, and this further improvement is described in Section 3.3.

### 3.3. Model Improvement

The previous section showed that the prediction error of the mass fraction factor exceeds the target value. This might be because the GM(1,1) modeling mechanism and model structure are difficult to obtain satisfactory fitting and prediction accuracy for if the original data are an approximate non-homogeneous sequence. Therefore, it is necessary to improve the modeling mechanism of the grey model to adapt it to the current system sequence.

The study found that the DNGM(1,1) (The DNGM (1,1) model is based on the whitening differential equation using parameters to Directly estimate the approximate Non-homogeneous sequence Grey prediction Model) model is suitable for approximate non-homogeneous data sequences. The model is based on the solution of the whitenization equation applicable to the approximate non-homogeneous model. Kramer’s law is then used to solve the corresponding coefficients, and the undetermined parameter values of the differential equations are obtained. A specific expression under the approximate non-homogeneous sequence-prediction model is obtained. Modeling avoids the error caused by directly using the difference equation to estimate the parameters, but it obtains a response function from the differential equation.

The DNGM(1,1) model is similar to the GM(1,1) modeling process except for the whitenization equation (shown in Equation (17)). The undetermined parameters in the equation are obtained from the intermediate parameters $\overset{\wedge }{a}={\left(\alpha ,\beta ,\gamma \right)}^{T}$.
(17)$$\frac{dx^{\left(1\right)}}{dt}+ax^{\left(1\right)}=bt+c$$

The intermediate parameters α, β and γ are obtained via Equations (18) and (19):(18)$$\overset{\wedge }{a}={\left(B^{T}B\right)}^{\left(-1\right)}B^{T}Y$$
(19)$$Y=\left[\begin{array}{c} x^{\left(1\right)}(2)\\ x^{\left(1\right)}(3)\\ \vdots \\ x^{\left(1\right)}(m) \end{array}\right]B=\left[\begin{array}{ccc} x^{\left(1\right)}(1) & 1 & 1\\ x^{\left(1\right)}(2) & 2 & 1\\ \vdots & \vdots & \vdots \\ x^{\left(1\right)}(m-1) & m-1 & 1 \end{array}\right]$$

On the basis of solving the intermediate parameters, the undetermined parameters a, b and c of the white differential equation are solved by Equation (20).
(20)$$\begin{array}{l} a=-\ln(\alpha )\\ b=\frac{a\beta }{1-\alpha }\\ c=\frac{a\gamma -b}{1-\alpha }+\frac{b}{a} \end{array}$$

We can then solve the response expression of the white differential equation. The adjacent two values of the responsive sequence are subtracted; finally, the analog value Equation (21) under the DNGM(1,1) model is obtained.
(21)$${\overset{\wedge }{x}}^{\left(0\right)}(k)=(1-e^{a})(x^{\left(0\right)}-\frac{b}{a}+\frac{b}{a^{2}}-\frac{c}{a})e^{-a(k-1)}+\frac{b}{a},k=1,2,\cdots m$$

The model error was evaluated using the average relative error MAPE. The corresponding Matlab program leads to the following in Table 4:

Table 4 shows that the fitting error of the mass fraction factor under the DNGM(1,1) model is 0.48%, and the prediction error is 1.62%. These conditions satisfy the expected target error requirement. The improved model is successful in predicting the diameter of the mass fraction factor.

In summary, under the GM(1,1) model, the average prediction error of the voltage factor is 0.81%, the average prediction error of the flow factor is 1.31%, and the average prediction error of the distance factor is 1.82%. Under the improved DNGM(1,1) model, the average prediction error of the mass fraction factor is 1.62%. The average prediction error of each factor is less than 3%. This indicates that the GM(1,1) model and the DNGM(1,1) model are successful in predicting the diameter of the electrospun nanofiber. Using any parameter within the allowable range of each factor, the corresponding nanofiber diameter can be accurately predicted.

## 4. Experimental Verification and Optimization of Multi-Variable Fiber Diameter Prediction

When multiple variables change simultaneously in actual production activities, the multi-variable grey model MGM(1,n) can be used to solve the fiber diameter prediction problem. To ensure the comparability of the electrospinning fiber diameter prediction experiment, the experiment needs to be carried out at the same formulation concentration. The concentration of the solution with a mass fraction of 12% is suitable, the surface of the fiber is smooth, and the fiber diameter distribution is uniform than other mass fractions. Therefore, the experimental conditions of a PAN mass fraction of 12% were selected to study the effect of the distance between the nozzle and the collector, the solution flow rate at the nozzle, and the loading voltage on the fiber diameter. Sequence numbers 1–6 in the following four-dimensional data were used for modeling, and sequence numbers 7–9 were used to test the prediction accuracy. The sequence of fiber diameter measurement values in Table 5 is recorded as $x_{1}^{\left(0\right)}$ (nm), the voltage factor sequence is recorded as $x_{2}^{\left(0\right)}$ (kV), and the distance between the head and the collector is recorded as $x_{3}^{\left(0\right)}$ (cm). The solution flow rate is reported as $x_{4}^{\left(0\right)}$ (mL/h). Table 5 records the data on the changes in fiber diameter as these three different factors change simultaneously.

The multi-variable grey model MGM(1,n) is an extension of the GM(1,1) model under the n variables. Similar to the GM(1,1) modeling process, only the main modeling part of MGM(1,n) is presented here. The first-order ordinary differential equations of the MGM(1,n) model are shown in Equation (22).
(22)$$\{\begin{array}{c} \frac{dx_{1}^{\left(1\right)}}{dt}=a_{11}x_{1}^{\left(1\right)}+a_{12}x_{2}^{\left(1\right)}+\cdots +a_{1n}x_{n}^{\left(1\right)}+b_{1}\\ \frac{dx_{2}^{\left(1\right)}}{dt}=a_{21}x_{1}^{\left(1\right)}+a_{22}x_{2}^{\left(1\right)}+\cdots +a_{2n}x_{n}^{\left(1\right)}+b_{2}\\ \vdots \\ \frac{dx_{n}^{\left(1\right)}}{dt}=a_{n1}x_{1}^{\left(1\right)}+a_{n2}x_{2}^{\left(1\right)}+\cdots +a_{nn}x_{n}^{\left(1\right)}+b_{n} \end{array}$$

In the MGM(1,n) model, n is the number of variables. The equation parameter used for solving and process transformation is similar to the GM(1,1) model and will not be described here. The sequence of analog values is expressed as Equation (23).
(23)$${\overset{\wedge }{X}}^{\left(1\right)}(k)=(X^{\left(1\right)}(1)+{\overset{\wedge }{A}}^{-1}\overset{\wedge }{B})e^{\overset{\wedge }{A}(k-1)}-{\overset{\wedge }{A}}^{-1}\overset{\wedge }{B}$$

The analog value sequence is inversely accumulated to obtain a predicted value corresponding to the original data. The accuracy of the model was evaluated by taking the average relative error MAPE.

Based on the multi-variable grey theory, the MGM(1,4) model was established and recorded as Model 1. We then used Matlab for solving and analyzing the data in Table 5 to get the relative error of fiber diameter fitting and prediction (Figure 3).

In Figure 3, Figure 4 and Figure 5, term B represents the relative error of the fit, and C represents the predicted relative error. Figure 3 shows that the maximum error is 41.33%. Further calculation shows that the average fitting error of the MGM(1,4) model is 8.62%, which does not meet the required accuracy requirements. Model improvements are needed to accommodate the current multivariate prediction of nanofiber diameter problems.

The selection of the background value coefficient α affects the sequence of system background values, thereby affecting the sequence of predicted values. The automatic optimization and weighting method is adopted to find the best background value coefficient from the internal modeling principle; this will improve the accuracy of the model.

The automatic optimization-weight method is as follows: the initial value of α is 0, and α increases by a small amount of Δα (Δα>0). We then identify the value of MAPE and select the value of α when the MAPE value is the smallest. This value is the best background value coefficient. The MGM(1,4) model that optimizes the background value coefficients is denoted as Model 2. Its relative error is shown in Figure 4.

Figure 4 shows that the accuracy of Model 2 is much higher than that of Model 1, and the average relative error has reached the expected target value. It satisfies the requirements of general quality.

These models are all integer-order models. In fact, the sequence of fractional-order properties is more common, and the fractional-order grey prediction model can better reveal the essence of system sequences [38]. In fact, in order to obtain better prediction accuracy, it is more effective to optimize the grey model via the fractional order principle.

The fractional-order optimized multi-variable grey prediction model has the following principle:$$X^{\left(0\right)}={\left\{X_{1}^{\left(0\right)},X_{2}^{\left(0\right)},\cdots ,X_{n}^{\left(0\right)}\right\}}^{T}=\left[\begin{array}{cccc} x_{1}^{\left(0\right)}(1) & x_{1}^{\left(0\right)}(2) & \cdots & x_{1}^{\left(0\right)}(M)\\ x_{2}^{\left(0\right)}(1) & x_{2}^{\left(0\right)}(2) & \cdots & x_{2}^{\left(0\right)}(M)\\ \vdots & \vdots & & \vdots \\ x_{n}^{\left(0\right)}(1) & x_{n}^{\left(0\right)}(2) & \cdots & x_{n}^{\left(0\right)}(M) \end{array}\right]$$

Here, $X^{\left(0\right)}$ is the original data sequence, $X^{\left(r\right)}$ is the r-order cumulative generation sequence of $X^{\left(0\right)}$, and the expression and production formula of $X^{\left(r\right)}$ as follows:(24)$$X^{\left(r\right)}=(x^{\left(r\right)}(1),x^{\left(r\right)}(2),\cdots ,x^{\left(r\right)}(n))$$
(25)$$x_{i}^{\left(r\right)}(k)=\sum_{j=1}^{k}\frac{\Gamma (r+k-j)}{\Gamma (k+1-j)\Gamma (r)}x_{i}^{\left(0\right)}(j)$$

k = 1, 2,…, *M*; i = 1, 2,…, *n*; Γ is a gamma function.

The differential equations of the fractional grey model FMGM(1,n) are expressed as:(26)$$\{\begin{array}{c} \frac{dx_{1}^{\left(r\right)}}{dt}=a_{11}x_{1}^{\left(r\right)}+a_{12}x_{2}^{\left(r\right)}+\cdots +a_{1n}x_{n}^{\left(r\right)}+b_{1}\\ \frac{dx_{2}^{\left(r\right)}}{dt}=a_{21}x_{1}^{\left(r\right)}+a_{22}x_{2}^{\left(r\right)}+\cdots +a_{2n}x_{n}^{\left(r\right)}+b_{2}\\ \vdots \\ \frac{dx_{n}^{\left(r\right)}}{dt}=a_{n1}x_{1}^{\left(r\right)}+a_{n2}x_{2}^{\left(r\right)}+\cdots +a_{nn}x_{n}^{\left(r\right)}+b_{n} \end{array}$$

The intermediate transformation and calculation process is similar to the MGM (1, n) model and will not be described here. The r-order sequence of FMGM(1,n) is shown in Equation (27).
(27)$${\overset{\wedge }{X}}^{\left(r\right)}(k)=(X^{\left(r\right)}(1)+{\overset{\wedge }{A}}^{-1}\overset{\wedge }{B})e^{\overset{\wedge }{A}(k-1)}-{\overset{\wedge }{A}}^{-1}\overset{\wedge }{B}$$

The simulation value of the original data can be generated by the r-order subtraction of Equation (27), and the accuracy of the model is evaluated by the average MAPE relative error.

It is known from Model 2 that selecting the appropriate background value coefficient α can improve the prediction accuracy. We next combined the fractional order and background value coefficient optimization as Model 3 and jointly explored a more accurate method. The steps are as follows:

Step 1: Set the initial value r=0 and accumulate the expression that generates $X^{\left(r\right)}$.

Step 2: Set the initial value of the background value coefficient α=0 and solve the model parameters by the least squares method.

Step 3: Solve the response function and further calculate the simulated value and the predicted value sequence.

Step 4: On the basis of 3, solve the average relative error MAPE.

Step 5: If r<2 and α<1, then add a small amount of Δα and Δr (Δr>0) and repeat the above operation.

Step 6: The corresponding r and α values when the MAPE is minimal is the optimization target value.

The above algorithm was written and calculated in Matlab. The average relative error MAPE is the smallest when the background value coefficient α is 0.5 and the fractional order is 1.006. The relative error probability map is shown in Figure 5.

Figure 5 shows that the further improved Model 3 has a large improvement in the fitting error and the prediction error, and the error is further reduced. The optimized model is closer to the actual sequence and can meet the needs of higher precision.

We next compared the prediction results of the original model and the improved model (Table 6). Model 1 is the original MGM(1,4) model, Model 2 is the MGM(1,4) model optimized for background value coefficients, and Model 3 is the MGM(1,4) model with background value coefficient optimization and fractional order combination.

The prediction error can be greatly reduced by selecting appropriate background-value coefficients during modeling (Table 6). The fractional order provides an effective way to improve the prediction accuracy. This makes the improved model more in line with the actual sequence. The combination of background value coefficient optimization and fractional order optimization can reveal the essence of the system sequence. The method has strong applicability and will be useful for predicting fiber diameter in electrospinning.

## 5. Conclusions

In this paper, a new prediction method was proposed in the field of electrospinning fiber-diameter prediction; the method was verified with experimental data. The GM(1,1) model exhibits high prediction accuracy related to single-factor variation on fiber diameter (e.g., loading voltage, solution flow rate, and receiving distance). The average prediction error is within 2%. The error of GM(1,1) in predicting the mass fraction factor sequence is 2.41%. The GM(1,1) prediction model under this factor is improved. The results show that the improved DNGM(1,1) model has a prediction error of 1.62%, and the accuracy improvement is obvious.

The average fitting error of the MGM(1,n) model was 8.62% in terms of the multifactorial influence on fiber diameter. The model was optimized via the background-value coefficient, and the average fitting error was reduced to 1.01%; the average prediction error was reduced to 1.33%, which greatly improves the prediction accuracy. The best background-value coefficients and orders are found by combining fractional-order optimization with background-value-coefficient optimization. The results show that the average prediction error of the further optimized model is only 0.38%. The final improved model shows higher accuracy and provides an effective model for predicting the diameter of the electrospun fibers.

Both the single-variable prediction model and the multi-variable prediction model achieve higher accuracy predictions with less sample data. The multivariate model can achieve effective auxiliary control of the fiber diameter by adjusting multi-factor parameters. This theory has important guiding significance in actual production activities.

This work combined background-value coefficient optimization and fractional-order optimization to enhance the applicability of the grey model in a study of a small number of samples and some unknown factors. At the same time, it is also a successful application of the grey prediction theory in the prediction of the diameter of electrospun fibers. This work adds to the field of application of grey prediction theory.

## Figures and Tables

**Figure 1 materials-12-02237-f001:**
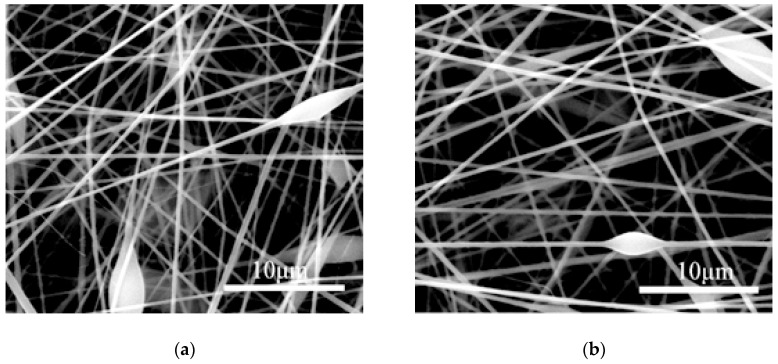
Bead-like structures at different voltages and PAN mass fraction of 12%. (**a**) Environmental scanning electron microscope (ESEM) image of bead-like structures at 10 kV; (**b**) ESEM image of bead-like structures at 11 kV. Scale bars: (**a**,**b**) 10 μm.

**Figure 2 materials-12-02237-f002:**
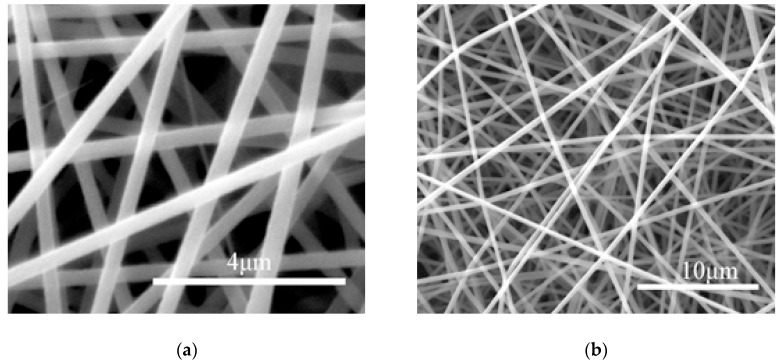
ESEM topography of electrospun fiber at different voltages and PAN mass fraction of 12%. (**a**) ESEM image of the electrospun fiber at 12 kV; (**b**) ESEM image of the electrospun fiber at 22 kV. Scale bars: (**a**) 4 μm; (**b**) 10 μm.

**Figure 3 materials-12-02237-f003:**
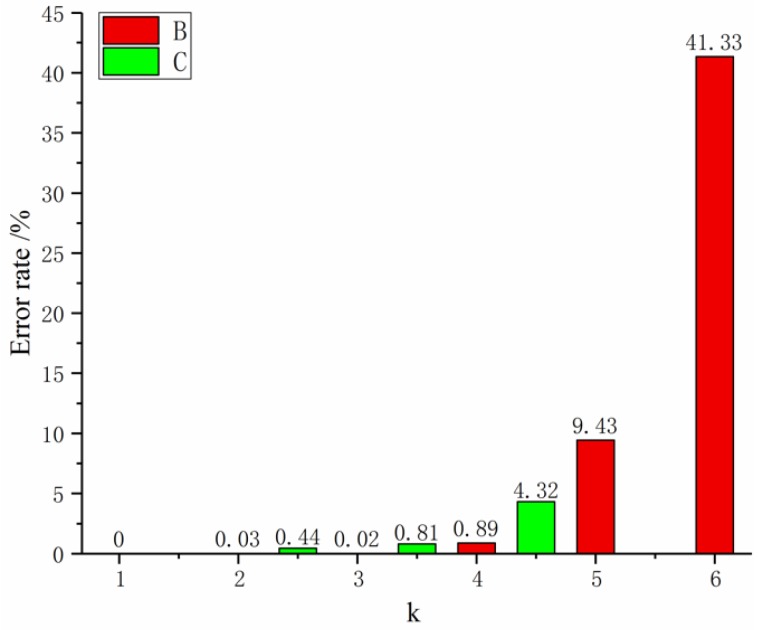
Model 1 relative error rate map. B: relative error of the fit; C: predicted relative error.

**Figure 4 materials-12-02237-f004:**
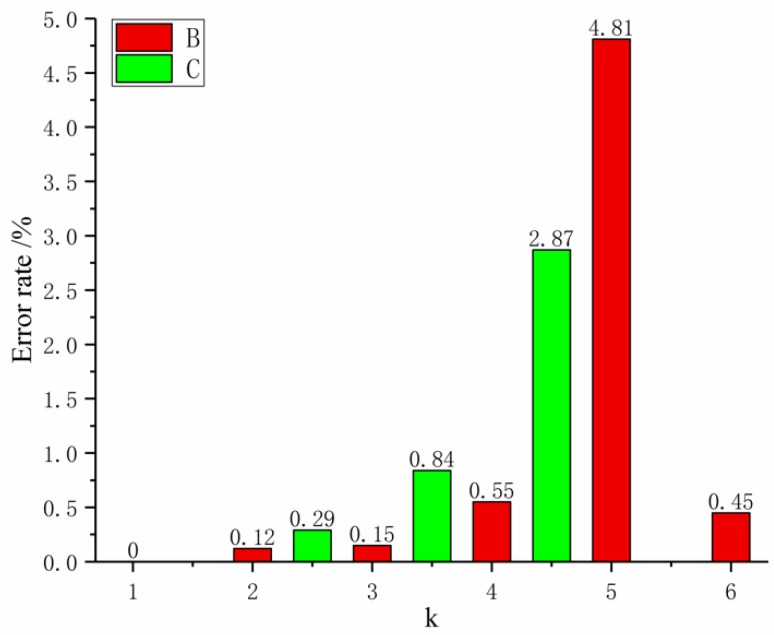
Model 2 relative error rate map.

**Figure 5 materials-12-02237-f005:**
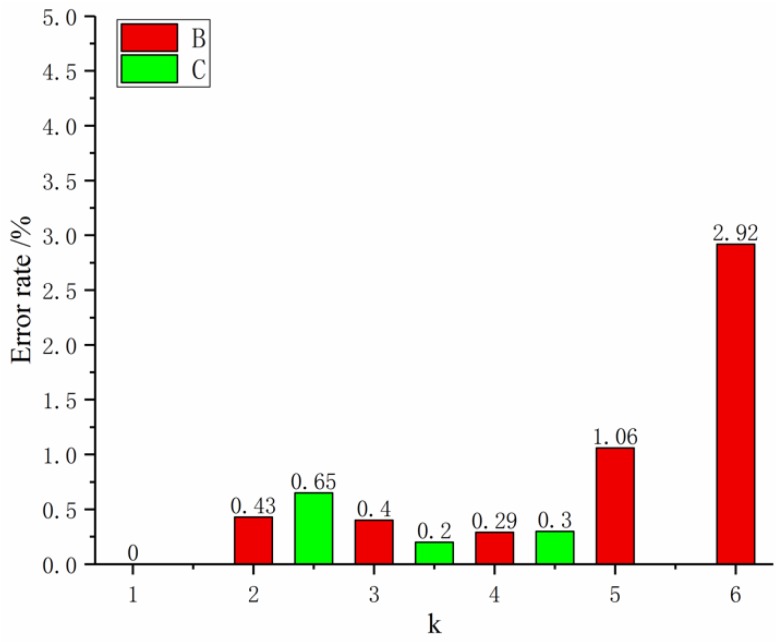
Relative error rate map of Model 3.

**Table 1 materials-12-02237-t001:** Experimental data sheet for the effect of individual factors on fiber diameter. L: the distance between the nozzle and the collecting plate; M: the mass fraction of polyacrylonitrile (PAN) solution; S: the flow rate of the solution at the nozzle; V: the loading voltage at the nozzle; $x_{i}^{\left(0\right)}$: the corresponding value of nanofiber diameter.

Number	V (kV)	$x_{1}^{\left(0\right)}$	M (%)	$x_{2}^{\left(0\right)}$	S (mL/h)	$x_{3}^{\left(0\right)}$	L (cm)	$x_{4}^{\left(0\right)}$
1	12	322.5	9	224.7	0.3	326.7	14	346.7
2	14	342.0	10	248.0	0.4	336.8	16	353.9
3	16	351.4	11	304.5	0.5	353.9	18	360.9
4	18	353.9	12	353.9	0.6	365.8	20	398.4
5	20	364.8	13	387.7	0.7	369.4	22	385.1
6	22	368.6	14	410.3	0.8	371.9	24	376.5
7	13	338.9	10.5	282.2	0.45	339.1	17	353.2
8	15	343.2	11.5	336.2	0.55	353.1	19	370.5
9	19	354.6	12.5	361.8	0.65	367.8	21	385.7

**Table 2 materials-12-02237-t002:** Voltage (V) and Mass Fraction (M).

Number	V (kV)	k	Real Value	Simulation Value	Simulation Error	Number	M (%)	k	Real Value	Simulation Value	Simulation Error
1	12	1	322.5	322.50	0.00%	1	9	1	224.7	224.70	0.00%
2	14	2	342	342.95	0.28%	2	10	2	248.0	266.53	7.47%
3	16	3	351.4	349.42	0.56%	3	11	3	304.5	299.37	1.69%
4	18	4	353.9	356.00	0.59%	4	12	4	353.9	336.25	4.99%
5	20	5	364.8	362.72	0.57%	5	13	5	387.7	377.67	2.59%
6	22	6	368.6	369.56	0.26%	6	14	6	410.3	424.20	3.39%
MAPE_1	-	-	-	-	0.38%	MAPE_1	-	-	-	-	3.35%
**-**	**-**	**-**	**-**	**Forecast Value**	**Forecast Error**	**-**	**-**	**-**	**-**	**Forecast Value**	**Forecast Error**
7	13	1.5	338.9	339.76	0.25%	7	10.5	2.5	282.2	282.47	0.09%
8	15	2.5	343.2	346.17	0.86%	8	11.5	3.5	336.2	317.27	5.63%
9	19	4.5	354.6	359.35	1.34%	9	12.5	4.5	361.8	356.36	1.50%
MAPE_2	-	-	-	-	0.81%	MAPE_2	-	-	-	-	2.41%

**Table 3 materials-12-02237-t003:** Flow Rate (S) and Receiving Distance (L).

Number	S (mL/h)	k	Real Value	Simulation Value	Simulation Error	Number	L (cm)	k	Real Value	Simulation Value	Simulation Error
1	0.3	1	326.7	326.70	0.00%	1	14	1	346.7	346.70	0.00%
2	0.4	2	336.8	342.79	1.78%	2	16	2	353.9	361.50	2.15%
3	0.5	3	353.9	350.97	0.83%	3	18	3	360.9	368.11	2.00%
4	0.6	4	365.8	359.35	1.76%	4	20	4	398.4	374.83	5.92%
5	0.7	5	369.4	367.92	0.40%	5	22	5	385.1	381.68	0.89%
6	0.8	6	371.9	376.70	1.29%	6	24	6	376.5	388.65	3.23%
MAPE_1	-	-	-	-	1.01%	MAPE_1	-	-	-	-	2.36%
**-**	**-**	**-**	**-**	**Forecast Value**	**Forecast Error**	**-**	**-**	**-**	**-**	**Forecast Value**	**Forecast Error**
7	0.45	2.5	339.1	346.86	2.29%	7	17	2.5	353.2	364.79	3.28%
8	0.55	3.5	353.1	355.13	0.57%	8	19	3.5	370.5	371.45	0.26%
9	0.65	4.5	367.8	363.91	1.06%	9	21	4.5	385.7	378.24	1.93%
MAPE_2	-	-	-	-	1.31%	MAPE_2	-	-	-	-	1.82%

**Table 4 materials-12-02237-t004:** Improved DNGM(1,1) model results.

Number	M (%)	k	Real Value	Simulation Value	Simulation Error
1	9	1	224.7	224.70	0.00%
2	10	2	248.0	246.99	0.41%
3	11	3	304.5	307.26	0.91%
4	12	4	353.9	352.41	0.42%
5	13	5	387.7	386.23	0.38%
6	14	6	410.3	411.56	0.31%
MAPE_1	-	-	-	-	0.48%
-	-	-	-	**Forecast Value**	**Forecast Error**
7	10.5	2.5	282.2	279.30	1.03%
8	11.5	3.5	336.2	331.46	1.41%
9	12.5	4.5	361.8	370.54	2.42%
MAPE_2	-	-	-	-	1.62%

**Table 5 materials-12-02237-t005:** Effect of simultaneous changes of multiple factors on fiber diameter.

Number	k	$x_{1}^{\left(0\right)}$	$x_{2}^{\left(0\right)}$	$x_{3}^{\left(0\right)}$	$x_{4}^{\left(0\right)}$
1	1	283.2	12	14	0.3
2	2	312.4	14	16	0.4
3	3	321.3	16	18	0.5
4	4	331.6	18	20	0.6
5	5	357.3	20	22	0.7
6	6	361.1	22	23	0.8
7	2.5	315.5	15	17	0.45
8	3.5	328.0	17	19	0.55
9	4.5	344.5	19	21	0.65

**Table 6 materials-12-02237-t006:** Comparison table of the three prediction results.

Number	Real Value	Model 1		Model 2		Model 3	
		Simulation Value	Simulation Error	Simulation Value	Simulation Error	Simulation Value	Simulation Error
1	283.2	283.20	0.00%	283.20	0.00%	283.20	0.00%
2	312.4	312.49	0.03%	312.03	0.12%	311.05	0.43%
3	321.3	321.23	0.02%	320.81	0.15%	320.00	0.40%
4	331.6	328.65	0.89%	329.78	0.55%	332.55	0.29%
5	357.3	323.60	9.43%	340.12	4.81%	353.51	1.06%
6	361.1	211.84	41.33%	359.49	0.45%	350.54	2.92%
MAPE_1	-	-	8.62%	-	1.01%	-	0.85%
**-**	**-**	**Forecast Value**	**Forecast Error**	**Forecast Value**	**Forecast Error**	**Forecast Value**	**Forecast Error**
7	315.5	316.89	0.44%	316.41	0.29%	317.54	0.65%
8	328	325.33	0.81%	325.24	0.84%	328.66	0.2%
9	344.5	329.63	4.32%	334.60	2.87%	345.52	0.30%
MAPE_2	-	-	1.86 %	-	1.33%	-	0.38%

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
