# Peer review of "Prediction and Optimization of Electrospun Polyacrylonitrile Fiber Diameter Based on Grey System Theory"

_materials, 2019, doi:10.3390/ma12142237_

Round 1
Reviewer 1 Report
General comments
The manuscript concerns the development of a new prediction method for the prediction of fiber diameter in electrospinning process.
The obtained results need to be deeper analyzed and comparison to experimental data has to be better highlight. Discussion of the obtained findings in needed to improve the quality of the manuscript. The final aim of the manuscript (i.e., why a prediction model could be useful in an electrospinning process?) has to be clarified
Some detailed comments are reported here below.
Detailed comments
- line 28: it depend on different electrospun parameters
- line 30: this sentence can be deleted
- lines 29-38: try to avoid repetitions
- line 45: “receiving plate”: usually it is named collector
- lines 47-48: do you mean Taylor’s cone
- line 59: clarify to what you are comparing these results
- lines 60-61: the two sentences are very similar. Revise them
- lines 68-70: you have already described it. Please avoid repetitions
- line 84: better explain to what you are referring to
- lines 60-95: after a general introduction to grey theory, you have to focus on the possible use of the model to predict fiber dimension on electrospun mats. In particular, you have to highlight if grey method has been previously proposed with the same aim of this manuscript
- line 96: “influencing factors”: detail on which factors you focused your investigation
- line 101: “the background value”: detail, in this case, to what you are referring
- lines 105-108: these are conclusions of your work. You have to move them in the Conclusion section
- line 112: this equipment does not need to be detailed
- line 115: “PAN powder”: you have to give a reason for using PAN as polymer for the electrospinning process
- line 120: “for labeling for use”: better explain what you mean by that
- lines 121-124: you have to specify if these parameters have been previously optimized. In that case, please add a reference
- line 125: give a reason for choosing those concentration values
- line 126: “other three factors”: specify to what you are referring
- line 127: specify to what you are referring
- lines 131-132: what do you mean by “was made into a sample”?
- lines 133-134: this is a result. You have to move it in the appropriate section
- Figure 1: you have to detail to which concentrations these images are referred; add the scale bar in the figure caption
- line 138: “the loading voltage”: you previously reported a final value for the voltage. You have to specify the values you considered
- line 140: you have to report the images of “bead-like structures”; you can add them as supplementary data
- line 149: “this”: it is not clear to what you are referring
- line 150: “this part”: give a reason for using only data in 7-9 for the model’s prediction accuracy
- line 156: “The four above groups of parameters”: detail the parameters you are referring to
- line 162: better explain what you mean by “are comparable”
- line 174: “other three groups of factors”, you have to report them in the supplementary data
- line 179: deeper explain
- line 190-191: you have to explain the reason for considering a value of alfa equal to 0.5
- line 208: you have to support your assessment by literature
- Table 2 and Table 3: you have to improve the comment to these data and you have to clarify if and for which parameters there is a better fitting between predicted values and measured values
- lines 257-259: deeper discuss this result
- lines 264-265: you have to better justify the reason for choosing 12% w/v PAN concentration
- give a reason and deeper discuss the reason for different accuracy for the different models here considered
- lines 322-323: better explain this point related to model 2
Reviewer 2 Report
The
prediction method proposed in this work is really interesting since the
theoretical value of the fiber diameter were verified by experimental
data. However the authors lack in motivating the effect of the
operating conditions of the electrospinning process on the fiber
diameter from an experimental and chemical-physical point of view. This
discussion will improve the scientific soundness of the work promoting
the prediction model as a method to design the properties of
electrospun fibers. By comparing Table 1 and Table 2, it is
evident that the voltage and the mass of polymer affect the average of
the simulation more than the distance and flow rate.Please discuss about
the effect of the operating conditions on simulation error. The
introduction must provide a more adequate background about the
electrospinning by discussing about the advantages and the challenges of
this technique. In the experimental section, authors are
suggested to provide information regarding model, company as well as the
production country for all the equipment used in this study.Please
provide further technical information about the polymer and the
solvent. Moreover authors should motivate the choice of the polymer. The
authors should provide info about the method used to measure the
diameter of the fiber and the experimental error of the measurements.
Round 2
Reviewer 1 Report
You have revised the manuscript improving the possible impact of your work. In the opinion of the reviewer, you have to add the comments to the reviewer's questions in the main text of your work, so that the results of your manuscript could be clearer.
You have to improve the figure labels. In fact, when you have two images, you have to clarify what is reported in each of them. Moreover, in the labels of the Figure where SEM images are reported you have to add the Scale bar
Reviewer 2 Report
The authors have improved the quality of the paper and the manuscript has presented in acceptable format.
I would encourage the authors to conduct a final grammatical review of the manuscript
